# Notational Analysis and Physiological and Metabolic Responses of Male Junior Badminton Match Play

**DOI:** 10.3390/sports11020035

**Published:** 2023-02-01

**Authors:** Ross Green, Andrew T. West, Mark E. T. Willems

**Affiliations:** Institute of Sport, Nursing and Allied Health, College Lane, University of Chichester, Chichester PO19 6PE, UK

**Keywords:** badminton, notational analysis, sport physiology, heart rate, lactate, respiratory exchange ratio, junior players, rally duration

## Abstract

We examined the game characteristics of badminton and the physiological and metabolic responses in highly trained male junior players. Players from a Badminton England accredited Performance Centre (n = 10, age: 14.0 ± 1.2 y, height: 1.69 ± 0.06 m, body mass: 59.1 ± 5.0 kg) completed a 20-m shuttle run test ($\dot{V}$O_2max_: 64 ± 7 mL·kg^−1^·min^−1^) and a simulated ability-matched competitive singles badminton game consisting of two 12-min games with a 2-min break wearing the COSMED K5 metabolic system with notational analysis. In five games, 427 points were contested with a rally time of 5.7 ± 3.7 s, a rest time of 11.2 ± 5.9 s, shots per rally of 5.6 ± 3.6, work density of 0.50 ± 0.21, an effective playing time of 32.3 ± 8.4%, and shots frequency of 1.04 ± 0.29. During badminton play, heart rate was 151 ± 12 b·min^−1^ (82 ± 10% of maximum heart rate), oxygen uptake was 39.2 ± 3.9 mL·kg^−1^·min^−1^ (62 ± 7% of $\dot{V}$O_2max_), and energy expenditure was 11.2 ± 1.1 kcal·min^−1^ with a post-game blood lactate of 3.33 ± 0.83 mmol·L^−1^. Compared to adult badminton play, the physiological responses of junior badminton are lower and may be due to the shorter rally durations. Male junior badminton players should be exposed to training methodologies which include rally durations in excess of what they encounter during match play so as to develop greater consistency. Our observations on game characteristics and physiological responses during junior badminton can be used to inform training practice.

## 1. Introduction

Badminton is a fast racket sport characterized by the ability to perform high-intensity and explosive movements such as lunges, jumps and overhead shuttlecock smashes interspersed with brief recovery periods [1]. Competitive success in badminton also depends on technical mastery, tactical shrewdness and mental toughness [2,3]. The sport was included in the 1992 Barcelona Olympics, with global participation in the sport in 2022 by over 300 million people [4]. Despite the popularity of badminton, there remains a lack of research on its physiological and metabolic characteristics, especially within competitive environments [5].

Notwithstanding the transformation to the scoring system in 2006, badminton is a sport that is perpetually evolving, as evidenced by research employing notational analysis [6,7]. Comparisons between men’s singles matches at the 2008 Beijing and 2012 London Olympics observed increases in a multitude of spatiotemporal parameters such as game duration, rally time, shots per rally and shots per game [8]. These modifications to the characteristics of badminton’s match-play explain why estimations of the bioenergetic requirements imposed during performance are highly disparate. For example, in a review by Phomsoupha et al. [9] in 2015, it was stated that between 60–70% of adenosine triphosphate resynthesis was provided by oxidative sources. However, newer quantifications have adjudged this figure to be in excess of 94% [10]. Accordingly, the applicability of antecedent research to contemporary performance may be limited, and without procuring specific knowledge of the current physiological demands, training regimens are likely outdated and ineffective [11].

Most studies on badminton have focused on senior badminton players [12,13,14]. However, junior athletes are not simply miniature adults, and their physiological responses are not commensurately scaled-down versions of adults [15,16,17]. For example, young tennis players incur a substantially greater energetic cost than their older counterparts [18]. Similarly, even when anthropometrically matched, age-related factors affect the physiological responses during cycling in participants of differing ages [19]. Consequently, if practitioners in badminton utilised the findings from research in adult players to develop training programs for youth players, this would be not only unsubstantiated but also likely erroneous. This restriction on the generalisability of findings to other participant demographics has been identified and addressed by researchers in a multitude of other sports. Specifically, the justification provided for investigations into juvenile table tennis and football was to identify diverging responses between senior and junior competitors [20,21]. These studies have resulted in the procurement and implementation of more systematic approaches towards long-term athlete development [22]. A prominent illustration of this transpires in tennis, whereby the equipment used by children is modified in order to prevent injuries and augment motor learning [23]. However, as these investigations have yet to be undertaken in badminton, training methodologies may not be developmentally appropriate [24]. This may contribute towards badminton’s high injury prevalence, with epidemiological studies documenting a preponderance of purportedly avoidable afflictions arising due to insufficient physical development [25].

Another potential limitation of the literature on badminton is presented by attempts to extrapolate findings from laboratory-based studies into realistic sporting environments. For example, the maximal oxygen uptake of badminton players is frequently obtained using incremental treadmill protocols [26,27]. However, laboratory testing is ineffective at replicating the sport-specific physiological and biomechanical particularities of badminton, such as the multidirectional movement patterns and muscular contractions that are undertaken on the court [28]. Therefore, the applicability of these methodologies to an intermittent sport with frequent changes of direction and fluctuations in movement velocity remains questionable [29]. Similarly, researchers attempting to conduct physiological profiling of racket sports often estimate pulmonary oxygen uptake indirectly under the assumption of a linear relationship with heart rate [30,31]. However, during badminton, this relationship has been shown to become uncoupled, and thus previous endeavours to establish the metabolic demands of match-play as a derivative of heart rate data are inaccurate and not representative of the true sporting demands [32]. The technological developments in portable metabolic devices may enhance the ecological validity of research, and they have been found to be comparable to the stationary metabolic cart and Douglas bag method, which are currently seen as the gold-standard for pulmonary gas exchange measurements [33,34]. As far as we know, portable metabolic devices have not been used on youth badminton players during simulated match play. In addition, only one study examined the physiological demands of badminton match play in youth elite male players (age: 16.0 ± 1.4 y old), but without observations on respiratory responses, and it is likely that some of these participants were nearing adulthood [35].

Therefore, the primary aim of the present study was to address the scarcity of knowledge on the physiological and metabolic characteristics of junior badminton match-play. The identification of the physiological attributes in a sport-specific context will allow the implementation of training strategies that are focused upon invoking these beneficial characteristics [36]. In addition, the quantification of work and recovery durations during intermittent sports has allowed practitioners to undertake training protocols with similar bioenergetic stimuli [37]. A secondary aim, therefore, was to use the notational analysis of badminton play to examine a potential link between the physiological responses and game characteristics in junior badminton play. It was hypothesized that notational analysis parameters would change during badminton in adolescent male badminton players. The observations of the present study may contribute to the development of evidence-based and age-appropriate training regimens for junior badminton players.

## 2. Materials and Methods

### 2.1. Participants and Ethical Approval

Ten highly trained male adolescent badminton players (age: 14.0 ± 1.2 y, height: 1.69 ± 0.06 m, body mass: 59.1 ± 5.0 kg) were recruited to take part in the study. Participants were members of a Badminton England accredited Performance Centre in the United Kingdom with at least bi-weekly group training sessions, individual coaching sessions and regular participation in Badminton England sanctioned tournaments. Self-reported weekly training hours were 8.7 ± 1.4 h with a training experience of 6 ± 1 y. At the time of study, participants ranged from age-group national champions to national ranking <30 in their age category. Participants completed a health history questionnaire and had no contraindicators to maximal exercise testing and musculoskeletal injury within the last six-months. Informed assent of the participants and informed consent of the parents/guardians was obtained. The study was approved by the Ethics Committee of the University of Chichester (code: 2106335).

### 2.2. Study Design

The study design was cross-sectional. Participants completed two testing sessions on separate days at the training facility. Participants were instructed to arrive for the testing having abstained from vigorous exercise for 48 h and being at least three h post prandial. To avoid the effects of diurnal variation, the testing was conducted at the same time (±2 h) each day [38].

### 2.3. Experimental Procedures

Before all testing, participants completed a standardised warm-up over a 20-m distance consisting of 4 × jogging, 4 × sidestepping, 1 × high knees, 1 × bum kicks, 1 × walking lunges with two steps between each lunge, 1 × leg swings with two steps between each leg swing and 1 × jogging with arm swings with a subsequent familiarisation to become accustomed to the equipment. Subsequently, participants underwent a 20-m shuttle run test (20 m MST) while wearing a portable metabolic device (K5, COSMED, Rome, Italy) (see below for details on use of the K5) to determine maximum oxygen uptake ($\dot{V}$O_2max_), maximum heart rate and peak blood lactate concentrations. Procedures for the 20 m MST were identical to those conducted by Ooi et al. [39], whereby participants travelled between two lines marked 20-m apart, in time to a gradually increasing ‘beep sound’. To maximise performance, strong standardised verbal encouragement was provided during the final stages and the test was terminated due to either volitional exhaustion or the failure to reach the line before the ‘beep’ on two successive occasions. At exercise termination, a fingertip capillary blood sample was taken at 1, 3 and 5-min and analysed for lactate (Lactate Pro 2, Arkray, Kyoto, Japan) with the highest value taken as peak blood lactate concentration. Match play intensity was expressed as a percentage of maximum heart rate during the 20 m MST (%HRM). The %HRM allowed for the quantification of the time spent in each of the intensity zones during game-play using cut-offs based upon previous research in badminton players [35]. Additionally, four criteria were used to ascertain maximal efforts during completion of the 20 m MST as used previously in youth racket sport athletes: (1) a plateau or increase of less than 1.5 mL·kg^−1^·min^−1^ in $\dot{V}$O_2_ despite an increase in exercise intensity; (2) a respiratory exchange ratio greater than 1.1; (3) a heart-rate above 95% of the age-predicted maximum value; and (4) a lactate concentration above 8 mmol·L^−1^ [40].

On another day, separated by a minimum of 72 h, participants competed in a singles match consisting of two 12-min games separated by a 2-min break. The duration of the match was informed by recent notational analysis conducted on players of a similar age and ability, whereas the duration of the break was based upon the time allowance permitted during Badminton England sanctioned tournaments [41]. Participants were matched for performance level (decided by author RG in capacity of extensive coaching experience with the cohort) and age, and informed that the results would be included in an internal ranking competition. During the two-minute break and immediately after the match, blood samples were taken from a fingertip of the non-dominant hand for lactate analysis. The notational analysis of the match-play was conducted by a qualified coach (RG) with over seven-years’ experience in accordance with procedures outlined by Faude et al. [42]. The notational measurements included rally time, rest time and shots per rally to allow the calculation of (1) work density = rally time/rest time, (2) effective playing time % = rally time/(rally time + rest time), and (3) shots frequency = number of shots per rally/rally time. Stopwatches were used for notational time measurements.

### 2.4. Measurements

During the experimental visits, participants were fitted with a heart rate monitor (Garmin HRM3, Olathe, KS, USA) and a portable metabolic device (K5, COSMED, Italy). Gas calibration was undertaken using ambient air and reference gas with a known composition (5% CO_2_, 16% O_2_). A CO_2_ scrubber was used to procure a 0% CO_2_ reading and a 3-litre syringe was used for flowmeter calibration (Hans Rudolf, Shawnee, KS, USA). Delay time was calibrated by fitting the participant with the mask and conducting several breaths at a predetermined rhythm. Pulmonary gas exchange measurements from the portable metabolic system were collected continuously throughout both tests using the breath-by-breath method, whereas heart rate data was recorded telemetrically [43].

### 2.5. Data Analysis

The breath-by-breath $\dot{V}$O_2_ data from the 20 m MST and match play were examined to exclude errant (i.e., non-physiological breaths caused by coughing or swallowing) data points using K5 software (Omnia, COSMED, Rome, Italy). Subsequently, a 30 s moving average was applied to gas exchange data during the 20 m MST and $\dot{V}$O_2max_ was determined as the highest $\dot{V}$O_2_ averaged over any 30 s period throughout the test. During match play, a 5 s moving average was calculated for both the gas exchange and heart rate data, as this sampling window reflects the temporal structure of badminton match play observed previously by Fernandez-Fernandez et al. [35] in players of a comparable background and in the present study. The respiratory exchange ratio was calculated as $\dot{V}$CO_2_/$\dot{V}$O_2_ in which $\dot{V}$CO_2_ is carbon dioxide production.

Energy expenditure was determined using the following Equations (1) and (2). Equation 1 is from Jeukendrup and Wallis [44].
(1)$$\mathrm{EE}\;(\mathrm{kcal}\cdot \min^{-1})=0.550\;\times \;\dot{V}{\mathrm{CO}}_{2}(L\cdot \min^{-1})\;+\;4.471\;\times \;\dot{V}O_{2}\;(L\cdot \min^{-1})$$

(2)
EE (kJ·min^−1^) = EE (kcal·min^−1^) × 4.184


Due to the small sample size, normality was assessed using the Shapiro-Wilk test [45,46]. Maximum oxygen data of the 20 m MST was normally distributed and a paired *t*-test was used to determine differences between measured $\dot{V}$O_2max_ and predicted $\dot{V}$O_2max_, whereas the Pearson product-moment correlation was used to investigate the relationship between measured $\dot{V}$O_2max_ and predicted $\dot{V}$O_2max_. To examine the differences from the notational analysis between the first and second game of the simulated match play and for the total game, a paired *t*-test was employed on normally distributed data (heart rate within 70–80%HRM, rally duration, rest time, work density and shot frequency), whereas a Wilcoxon signed-rank test was used on non-normally distributed data (shots per rally, effective playing time and the remaining heart rate data). Statistical testing was conducted on GraphPad Prism (Prism 5 for Windows) with the significance set at *p* < 0.05. *p*-Values of 0.05 ≤ *p* ≤ 0.1 were interpreted according to guidelines by Curran-Everett and Benos [47] in that there may be a true effect for a difference. Cohens’ d effect sizes were calculated and interpreted as small: 0.2 ≤ d < 0.5; moderate: 0.5 ≤ d ≤ 0.79 and large: d ≥ 0.8). Data are presented as mean ± SD, range and 95% confidence intervals.

## 3. Results

### 3.1. 20 m MST and Physiological Responses

The duration of the 20 m MST test was 580.9 ± 75.3 s (range: 489.0–685.0 s, 95% CI [527.0, 634.8]), with participants completing 85.3 ± 13.8 20 m shuttles (range: 69–104 shuttles, 95% CI [75.4, 95.2 shuttles]). Peak values for blood lactate concentration, heart rate and respiratory exchange ratio were 8.0 ± 0.8 mmol·L^−1^ (range: 6.9–9.6 mmol·L^−1^, 95% CI [7.5, 8.6 mmol·L^−1^]; 185 ± 9 b·min^−1^ (range: 163–194 b·min^−1^, 95% CI [179, 192 b·min^−1^] and 0.99 ± 0.07 (range: 0.90–1.10, 95% CI [0.94, 1.04]). The measured $\dot{V}$O_2max_ of 64 ± 7 mL·kg^−1^·min^−1^ (range: 51–79 mL·kg^−1^·min^−1^, 95% CI [58, 69 mL·kg^−1^·min^−1^]) was 13% higher (*p* = 0.003) than the predicted $\dot{V}$O_2_max of 56 ± 4 mL·kg^−1^·min^−1^ (range: 52–62 mL·kg^−1^·min^−1^, 95% CI [54, 59 mL·kg^−1^·min^−1^]) with the equation from Léger et al. [48]. There was a significant correlation coefficient for the relationship between the measured $\dot{V}$O_2max_ and predicted $\dot{V}$O_2_max (*r* = 0.65, *p* = 0.043).

### 3.2. Match Play: Notational Analysis

Five badminton matches of 2 × 12-min were analysed with a total of 427 points. The five-match characteristics of the 2 × 12-min badminton games are presented in Table 1. Between the first and second game, there was a 15% increase in rally duration (Table 1, *p* = 0.005). For the first and second game, the 95% CI for rally duration were [4.8, 5.7 s] and [5.6, 6.6 s]. In the second game, the shots per rally were higher (Table 1, *p* = 0.012). For the first and second game, the 95% CI for shots per rally were [4.8, 5.8] and [5.4, 6.4]. In the second game, the work density was higher (Table 1, *p* < 0.001). For the first and second game, the 95% CI for work density was [0.45, 0.51] and [0.51, 0.55]. In the second game, the effective playing time (%) was higher (Table 1, *p* < 0.001). For the first and second game, the 95% CI for effective playing time (%) were [29.7, 32.1%] and [32.8, 34.9%]. Between the first and second game, there may have been a true effect for a difference for shots frequency (*p* = 0.093). For the first and second game, the 95% CI for shots frequency were [1.03, 1.11] and [0.97, 1.05]. Only points played in the second game showed a moderate effect size.

For rally duration of the badminton games, 86% of rallies were completed within 10 s, with rallies between 4 and 6 s being the most common overall (Figure 1). Similarly, 67% of rallies recommenced within 12 s, with only 13% of rest time between rallies exceeding 18 s (Figure 2).

### 3.3. Match Play: Physiological and Metabolic Responsess

The observations for the physiological and metabolic responses during match play are provided in Table 2.

Heart rate during match play was mostly situated between 70–80% of the maximum heart rate (Figure 3). It seems that the heart rate was situated in the higher zones during the first 12-min game and the lower zones during the second 12-min game.

## 4. Discussion

The aim of the present study was to provide notational analysis observations as well as physiological and metabolic responses during 2 × 12-min badminton play in a novel cohort of adolescent highly trained male players (age: 14.0 ± 1.2 y). The cardiorespiratory and metabolic demands were high but seem to be lower than those previously documented in highly trained adult players, potentially due to shorter rally durations. The low blood lactate concentrations were similar to those of more experienced players, providing further evidence that the aerobic and phosphagen energy systems are of great importance during badminton match play.

### 4.1. Maximal Endurance Test–the 20 m MST

The 20 m MST test was chosen to assess aerobic capacity due to its greater applicability to intermittent sports with frequent changes in direction and speed [49]. It has high to very high test-retest reliability and is recommended for use with adolescents [50]. The $\dot{V}$O_2max_ in the present study was similar to the $\dot{V}$O_2max_ of 64.6 ± 4.3 mL·kg^−1^·min^−1^ in elite male junior badminton players (17.2 ± 1.2 y) [51] and higher than the $\dot{V}$O_2max_ of 45.2 ± 8.7 mL·kg^−1^·min^−1^ in recreational badminton players [13]. It is likely that training-induced adaptations have contributed to the high level of cardiorespiratory fitness seen in the cohort of the present study [52], but a genetic predisposition cannot be excluded. The crucial aspects of badminton match play (e.g., overhead jump smashes) are metabolically dependent upon the phosphagen system, with the resynthesis of phosphocreatine being an oxidative process that is superimposed on top of largely aerobic energy provision [53]. Consequently, high aerobic capacity would improve a player’s ability to recover from strenuous rallies [54]. In the present study, the regression equation used by Léger et al. [48] underestimated the $\dot{V}$O_2max_ measured with the K5 COSMED, as has been reported before [49]. During the 20 m MST, the deceleration, change of direction, and acceleration requires anaerobic metabolism that has a meaningful contribution to the outcome of the test [55]. Therefore, a strength of the present study was the use of the portable metabolic system K5 COSMED during the 20 m MST to allow physiological match responses as a percentage of the measured $\dot{V}$O_2max_.

### 4.2. Match Play Characteristics–Notational Analysis

In the present study, match duration (i.e., 2 × 12-min) was informed by recent notational analysis involving participants with similar characteristics. However, it seems that this time period was too long for a two-set match, as ability-matched participants would frequently go to the ‘two-points clear’ situation with players inseparable up until 20 points each. This discrepancy may be attributable to the contextual differences between the present study and that of Rojas-Valverde et al. [41]. In Rojas-Valverde et al. [41], the matches were part of an international badminton competition with match referees enforcing Badminton World Federation rules including a 1-min drinks break when one competitor reaches 11-points in each game. Conversely, the participants in the present study were encouraged to play in accordance with Badminton England rules for junior competition. Therefore, the drinks break was not enforced and all participants opted not to take the break, matching the approach they would employ during competitive tournaments. Although this ensured that the matches were representative of the normal competitive demands, the participants in the present study played up to two minutes longer than those in the study by Rojas-Valverde et al. [41]. In addition to this, all other values for the match play characteristics were mostly similar to observations in the literature. Rally duration (i.e., 5.7 ± 3.7 s) was more prolonged than the 4.62 ± 0.86 s in less trained male Malaysian competitors (age: 16.1 ± 0.8 y) [56] and substantially lower than the rally duration of 10.4 ± 2.1 s in men’s singles matches at the 2012 Olympics [57]. Rally duration is determined by the players’ ability to keep the shuttle within the confines of the court and the ability to hit winning shots with a minimal amount of unforced errors being more consequential to a successful outcome in a competitive match [58]. As a result, training may consist of technical practices intended to enhance consistency and provide individuals with the ability and adaptability to play more sophisticated shots such as top-spins and slices [59]. Therefore, as improved competency is typically accompanied by an increase in rally duration, the rally duration in the present study is indicative of the participants’ highly trained but immature status [60]. Similarly, the duration of rest periods was longer than it would be for players of a lower ability [56]. Strong positive correlations between rally time and rest time indicates that an increased work period elicits greater homeostatic perturbations to recover from [61]. The insufficient elongation of rest periods after prolonged rallies results in an increased rate of unforced errors [62]. Consequently, players in the present study may have a spent longer time at low intensities to allow the resynthesis of phosphocreatine and the restoration of the intracellular milieu to enhance their readiness for the next point [63]. In elite players, the strength of the relationship between rally and rest durations is diminished, suggesting that recovery is not the only factor mediating rest periods in this demographic [7]. It has been proposed that this period represents a time in which elite players strategically plan the next point [11]. The importance of this has been magnified by the higher shuttlecock velocities and thus the effectiveness of the smash shot, which has resulted in players competing for control of the front-court in order to ascertain a shot with an upwards trajectory from the opponent [64]. Therefore, the longer rest durations in elite players may represent greater tactical planning combined with increased physiological demands. Similar findings have been uncovered in tennis, whereby more experienced players take longer between points in order to generate more complex strategies through the diagnostic evaluation of previous points and the prediction of future stylistic alterations of the opponent [65]. This disproportionate increase in rest time compared to rally time in elite adult competitors explains why the values for work density and effective playing time are higher in the present study than they are in most other studies. For example, work density during the first and second set of Olympic matches was 0.38 and 0.36 [6]. Similarly, effective playing time in senior international matches was 31.2% [42]. In other racket sports, comparable results with sub-elite players documented a higher effective playing time than elite players e.g., [66]. Our observations for shot frequency are in agreement with the results of 1.03 that were found in Malaysian state level males [56], but lower than during elite male singles matches [6]. A longitudinal analysis of Olympic matches over 20-years showed perpetual increases in shot frequency amounting to 34%, suggesting that badminton continues to evolve towards faster exchanges between players [7]. Consequently, advice for adolescent players could be to develop the physical and anticipatory skills required to engage in more explosive rallies [64,67].

### 4.3. Match Play–Physiological Characteristics

Heart rate monitoring allows for the assessment of cardiovascular strain during sporting performance and can be used to monitor exercise intensity and quantify internal training load [68]. The average heart rate of the players in the present study was 151 ± 12 b·min^−1^, corresponding to 82 ± 10%HRM. Despite lower average heart rates in our study, similar values of 83%HRM were reported in sub-elite young adult badminton players [10]. Elite youth competitors had higher average heart rates of 183 b·min^−1^ and %HRM of 89.8% during the first game and 184 b·min^−1^ and 90.8%HRM during the second game of simulated match play [35]. Similarly, the average %HRM in elite adult players often exceeds 90% during competitive match play [42,58]. This would suggest that match play in the present study invoked less cardiovascular strain than previously documented in players of a greater ability. However, it has been postulated that the time spent in various heart rate zones provides more meaningful information on the demands of match play due to the intermittency of work periods [35]. In the present study, heart rate was mostly between 70–80%HRM with a trend towards a greater time spent in the lower heart rate zones during the second game despite significantly longer rally durations. This is in contrast to previous research which found that highly trained adults spend over half of the total playing time with heart rates in the 90–100%HRM zone with a shift towards higher heart rates in the latter games [35,64]. Although no substantiating data was collected, a potential reason for this may be related to the emotional responses experienced during the test. Despite undertaking familiarisation periods before each test, participants still reported feelings of nervousness after match play. It is possible that increased psychological arousal during the early stages of match play mediated higher sympathetic activation and thus the greater secretion of catecholamines [69] and increasing heart rate irrespective to the demands of the exercise. Longer familiarisation may be required to fully accustom young players to the portable metabolic device.

As far as we know, this is the first study that measured the oxygen uptake of youth players during simulated badminton match play. Match play resulted in an average $\dot{V}$O_2_ of 39.2 ± 3.9 mL·kg^−1^·min^−1^, corresponding to a metabolic equivalent of 11.2 and a classification of vigorous intensity exercise [70]. Our value was higher than the 34.4 ± 5.8 mL·kg^−1^·min^−1^ in recreational badminton matches and lower than the 46.0 ± 4.5 mL·kg^−1^·min^−1^ in elite adult badminton players [13,42]. However, relative $\dot{V}$O_2max_ in the present study was 62% and substantially lower than the 76% and 73% in recreational and elite adult badminton players [13,42]. Although the lower values compared to the untrained athletes may be due to the lower $\dot{V}$O_2max_ (i.e., 45.2 ± 8.7 mL·kg^−1^·min^−1^) in Deka et al. [13], the $\dot{V}$O_2peak_ in Faude et al. [42] was similar to that of the current study, suggesting that the players were exercising at higher relative intensities.

Consequently, the heart rate and oxygen uptake data suggest that the physiological demands of junior badminton match play are substantially lower than those previously documented in highly trained adults [64]. As the physiological load is influenced by the duration of work periods, the shorter rally duration in the present study appears to elicit diminished homeostatic perturbations [61]. However, a considerable amount of time is still spent at very high intensities (16.2% of total match duration at >90%HRM). At these intensities, an individual’s fitness can be a decisive factor towards the outcome of the match [71]. Therefore, for adolescent badminton players, training strategies should aim to enhance the ability to maintain technical proficiency during periods of heightened physiological strain.

In the present study, badminton play by highly trained male adolescents had an energy requirement equivalent to ~670 kcal·h^−1^ and was comparable to the 660 kcal·h^−1^ in racquetball in a mixed cohort [72], but higher than the ~600 kcal·hour^−1^ seen during male tennis matches [73]. The respiratory exchange ratio in the present study was lower than the 0.99 recorded during adult badminton match play [42] and comparable to the junior table tennis ratio of 0.86 [74]. However, the lack of steady-state conditions during racket sports prohibits definitive statements regarding substrate utilisation [75]. The low lactate concentrations, however, suggests that energy production through anaerobic glycolysis was limited. It is likely that the physiological adaptations elicited by frequent engagement in badminton training effectuate an amelioration in oxidative capacity and lactate clearance, thus delaying or even preventing the occurrence of metabolic acidosis [76], and includes increases in mitochondrial volume and density, oxidative enzyme activity and capillarization. However, it needs to be noted that the lactate observations are only reflective of the glycolytic activity during the few minutes preceding the sample and are unlikely to determine the true contribution of the anaerobic breakdown of glucose.

### 4.4. Limitations

Portable metabolic devices facilitate pulmonary gas exchange measurements during training matches, but their use during official tournaments is prohibited. Therefore, it remains unknown how accurately simulated matches replicate high pressure competitive conditions, which could alter observations for notational analysis and physiological responses. In addition, it is highly likely that the wearing of a portable metabolic device results in lower playing quality and exercise intensity than during normal match play [42]. This is especially pertinent in the demographic of the present study, as these devices are larger relative to their current anthropometric characteristics. It should also be noted that the biological maturation status of the participants was not determined, which is important in talent selection programs [77]. In addition, our observations should not be generalized to internationally competitive adolescent male badminton players.

Lastly, notational analysis was undertaken to provide insight into the underlying causes of the physiological responses. It was employed using oversimplifications so that different actions could be easily categorised. For example, the use of ‘shots per rally’ in the present study fails to differentiate between shots that invoke substantially differing metabolic requirements. The execution of a highly demanding movement such as the jump smash would result in greater energetic demands than a simple net shot [78]. Therefore, future research should aim to utilise more objective methodologies such as local positioning systems and accelerometers to quantify external load. A larger sample size may have provided significant observations for shot frequency and rest time between the games. As far as we know, only the study by Abdullahi et al. [12] established the relationships between measures of internal and external load in badminton. However, internal load was only quantified using the heart rate, which has been known to underestimate sporting demands due to the delayed response to high intensity actions [79]. Consequently, a wider assortment of internal load parameters should be investigated in combination with external load ones. Developing knowledge of the relationship between the sporting demands and individual adaptive responses will allow practitioners to perform load monitoring and design training regimens more effectively.

## 5. Conclusions

Observations in the present study in highly trained male adolescent badminton players corroborate previous research that characterises badminton as a sport with relatively high cardiovascular and metabolic demands. However, the intensity of match play in male juniors appears to be lower than that of their highly trained adult counterparts. This is likely due to the lower technical competency of the adolescent population, which results in earlier rally cessation. Consequently, the cardiorespiratory fitness of the male adolescent athletes in the present study is not likely to be a limiting factor with regard to performance levels, and more attention should be paid to motor skill acquisition. To this end, badminton-specific training drills such as the ‘multi-shuttle drill’ may be an effective and time-efficient training modality to simultaneously provide the stimulus required to maintain sufficient aerobic fitness while developing technical and tactical astuteness.

## Figures and Tables

**Figure 1 sports-11-00035-f001:**
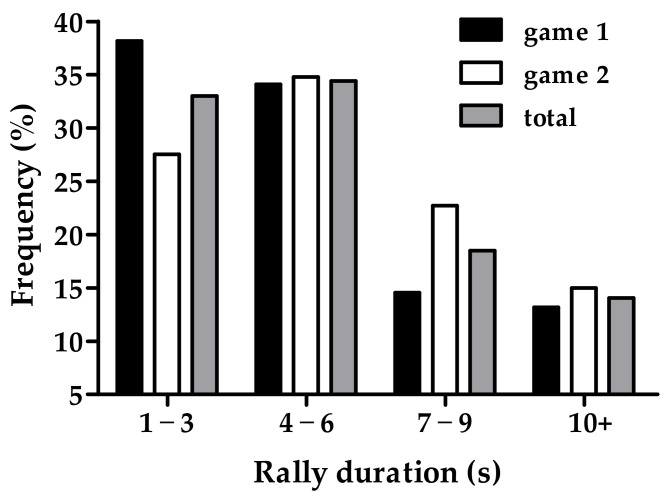
Frequency distribution of the rally duration for the first and second 12-min game and for the game in total.

**Figure 2 sports-11-00035-f002:**
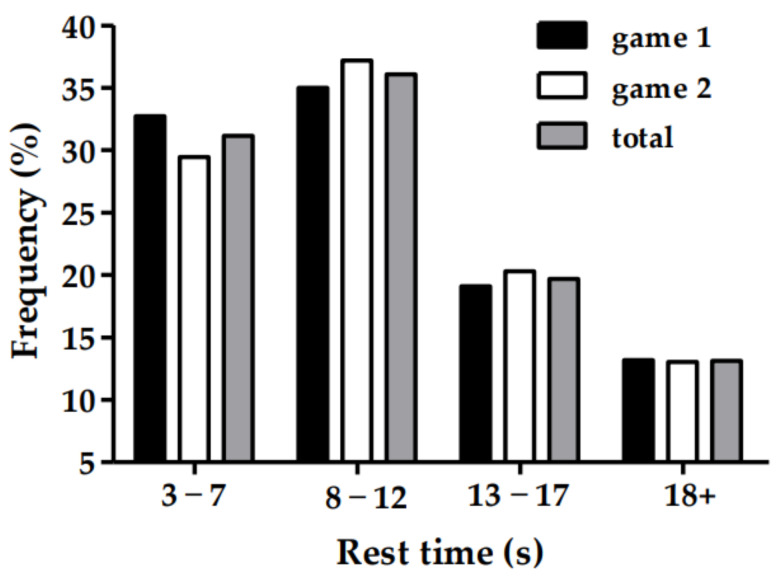
Frequency distribution of the rest time between rallies for the first and second 12-min half and for the game in total.

**Figure 3 sports-11-00035-f003:**
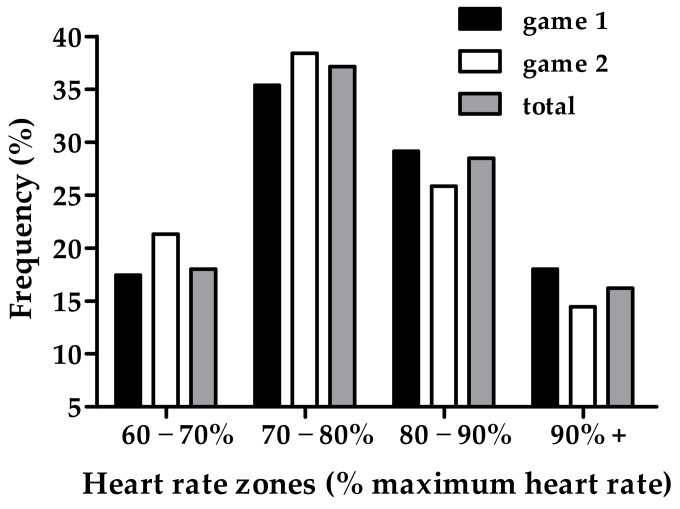
Frequency distribution of the heart rate zone as a percentage of maximum heart rate in the first and second 12-min half and for the game in total.

**Table 1 sports-11-00035-t001:** Badminton match characteristics of junior male players of a 2 × 12-min game with a 2-min break. Data (mean ± SD) of five 2 × 12-min games.

Parameter	1st Game	2nd Game	Effect Size	Total
Points played	44 ± 4	41 ± 5	−0.66	84 ± 6
Rally duration (s)	5.3 ± 3.5	6.1 ± 3.9 *	0.22	5.7 ± 3.7
Rest time (s)	11.1 ± 5.8	11.3 ± 5.9	0.03	11.2 ± 5.9
Shots per rally	5.3 ± 3.7	5.9 ± 3.5 *	0.17	5.6 ± 3.6
Work density	0.48 ± 0.23	0.53 ± 0.17 *	0.25	0.50 ± 0.21
Effective playing time (%)	30.9 ± 9.0	33.8 ± 7.4 *	0.35	32.3 ± 8.4
Shots frequency	1.07 ± 0.31	1.01 ± 0.27 ^$^	−0.21	1.04 ± 0.29

*, indicates a difference between the first and second game. ^$^, indicates that there may be a true effect for a difference. Work density = rally time/rest time. Effective playing time (%) = rally time/(rally time + rest time). Shots frequency = number of shots per rally/rally time.

**Table 2 sports-11-00035-t002:** Physiological and metabolic responses of the 2 × 12-min badminton games. Data are mean ± SD, 95% CI and range values.

Parameter	Mean ± SD	95% CI	Range
Heart rate (beats·min^−1^)	151 ± 12	[142, 160]	133–169
Heart rate (%HRM)	82 ± 10	[75, 90]	69–101
Lactate during break (mmol·L^−1^)	2.42 ± 0.44	[2.10, 2.73]	1.90–3.19
Lactate post-match (mmol·L^−1^)	3.33 ± 0.83	[2.74, 3.93]	1.60–4.50
$\dot{V}$O_2_ (ml·kg^−1^·min^−1^)	39.2 ± 3.9	[36.5, 42.0]	34.0–44.8
$\dot{V}$ (% of $\dot{V}$O_2_max)	62 ± 7	[57, 67]	54–80
Respiratory exchange ratio	0.84 ± 0.07	[0.79, 0.89]	0.77–1.00
Minute ventilation (L·min^−1^)	54.1 ± 4.4	[51.0, 57.3]	46.9–62.2
Energy expenditure (kcal·min^−1^)	11.2 ± 1.0	[10.4, 11.9]	10.0–12.7
Energy expenditure (kJ·min^−1^)	46.7 ± 4.4	[43.6, 49.9]	41.8–53.2
Total energy expenditure (kcal)	268 ± 25	[250, 286]	240–305

%HRM, percentage of maximal heart rate.

## Data Availability

Data can be obtained on reasonable request.

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
