# Peer review of "Notational Analysis and Physiological and Metabolic Responses of Male Junior Badminton Match Play"

_sports, 2023, doi:10.3390/sports11020035_

Round 1

Reviewer 1 Report

Dear Authors,

Thank you for giving me the opportunity to review this study.

In this article, you proposed a notational Analysis and Physiological and Metabolic Responses of Junior Badminton Match Play. The article is of scientific interest, but there are some issues that have to been addressed.

In the participants section, the years of experience of the players should be included in the characteristics of the sample.

In the method I am curious to know why, given all the favorable conditions to be able to analyze complete matches, it was only decided to carry out Five badminton matches of 2 x 12-min and the possibility of making it more "real" was not contemplated, the matches taking place until the end, since there could be more intense games that could modify, for example, the peak values ​​of lactate or VO2.

As a final suggestion, I think that some conclusion could be included more at a practical level than a better response to the secondary aim, therefore, was to use notational analysis of badminton play to examine a potential link between the physiological responses and game characteristics in junior badminton play. The observations of the present study may contribute to the development of evidence-based and age-appropriate training regimens for junior badminton players.

As limitations of the study, it should be noted that the results obtained must be taken with care as the sample size is very small. Also that the results would only be for male players since there is no female data.

Reviewer 2 Report

General comments to the authors

This study aimed to investigate the game characteristics of badminton and the physiological and metabolic responses in highly trained male junior players.

In this study, using the gold standard k5 portable metabolic devices system made the study very powerful. But It is possible that using a portable metabolic gadget will result in lower playing standards and workout intensity than when using it for regular match play. Despite this adverse situation, it has been a very high-quality study to determine the match performances of young badminton players. However, some parts of the study need improvement.

Introduction

This section is well designed and well-written.

Line 99: Add an hypothesis.

Materials and Methods

Line 101: Participants

Explain more about adolescent badminton players' game levels. (number of tournaments played "national - international", number of matches, ranking of players)

Line 147: How was the performance level of the participants determined?

Were the maturation levels of the participants determined? if not measured, add to the limitation.

Line 151: Are the matches recorded with a camera?

which analysis program was used (e.g. kinovea program)?

write down the camera model, the number of cameras, and where it is located in the field.

Line 422: add to limitation

            A limited number of participants and only male

             not knowing the game levels of the subjects,

Line 175: Rewrite the data analysis section more simply. (removable between 171 and 191)

Line 233: Add the effect size to the table for a clearer understanding of the comparison of the 1st and 2nd game in match analysis

Reviewer 3 Report

The primary aim of the present study was to knowledge on the physiological and metabolic characteristics of junior badminton match-play. A secondary aim, therefore, was to use notational analysis of badminton play to examine a potential link between the physiological responses and game characteristics in junior badminton play.

It is a very practical research and a well-written manuscript, although it raises some doubts, especially methodological ones.

Here are my contributions:

Introduction

  • Line 51: Is there not an example/reference of a sport modality that is more similar to the physiological requirements of badminton? (tennis, paddle…)
  • The priority objective cannot be the use of a material/instrument. Knowing the physiological and metabolic characteristics of badminton matches in the lower categories would be the main objective.

Methodology

  • What has been the best competitive result obtained by the participants?
  • Line 125: It is not clear in the methodology how the shuttle run test (20m MST) is performed. Was the test performed with a gas analyzer or was the peak oxygen consumption estimated? In the lactate test, the meter used is specified, here it is not stated….. Having a gas analyzer why was not performed a test with the analyzer? with is the objective to evaluate the peak oxygen uptake and not the maximal oxygen uptake?
  • Instead of playing 2x12min why didn't they play a "real" game?
  • Line 214, how can VO2peak be 13% higher than the estimated VO2max?

Discussion

- Don't you think that the level of the participants may be a limitation of the study when compared to existing literature?

Round 2

Reviewer 3 Report

The authors have responded satisfactorily to the contributions and comments made. Congratulations.